# Comparison of the complications between minimally invasive surgery and open surgical treatments for early-stage cervical cancer: A systematic review and meta-analysis

Yilin Li[1,2☯], Qingduo Kong[1,2☯], Hongyi Wei[2], Yongjun Wang[2]*

**1** Clinical Medical College, Weifang Medical University, Weicheng District, Weifang, Shandong, China,
**2** Division of Gynecology and Obstetrics, Life Science Park of Zhongguancun, Peking University International Hospital, Changping District, Beijing, China

☯ These authors contributed equally to this work.
* wyongjunhys@sina.com

**Data Availability Statement:** All relevant data are within the paper and its Supporting Information files.

## Abstract

### Background

This meta-analysis comprehensively compared intraoperative and postoperative complications between minimally invasive surgery (MIS) and laparotomy in the management of cervical cancer. Even though the advantages of laparotomy over MIS in disease-free survival and overall survival for management of gynecological diseases have been cited in the literature, there is a lack of substantial evidence of the advantage of one surgical modality over another, and it is uncertain whether MIS is justifiable in terms of safety and efficacy.

### Methods

In this meta-analysis, the studies were abstracted that the outcomes of complications to compare MIS (laparoscopic or robot-assisted) and open radical hysterectomy in patients with early-stage (International Federation of Gynecology and Obstetrics classification stage IA1-IIB) cervical cancer. The primary outcomes were intraoperative overall complications, as well as postoperative aggregate complications. Secondary outcomes included the individual complications. Two investigators independently performed the screening and data extraction. All articles that met the eligibility criteria were included in this meta-analysis.

### Results

The meta-analysis finally included 39 non-randomized studies and 1 randomized controlled trial (8 studies were conducted on robotic radical hysterectomy (RRH) vs open radical hysterectomy (ORH), 27 studies were conducted on laparoscopic radical hysterectomy (LRH) vs ORH, and 5 studies were conducted on all three approaches). Pooled analyses showed that MIS was associated with higher risk of intraoperative overall complications (OR = 1.41, 95% CI = 1.07–1.86, P<0.05) in comparison with ORH. However, compared to ORH, MIS was associated with significantly lower risk of postoperative aggregate complications (OR =

**Funding:** This article was funded by "Capital's Funds for Health Improvement and Research" (2020-2-8022). The funders had no role in study design, data collection and analysis, decision to publish, or preparation of the manuscript.

**Competing interests:** The authors have declared that no competing interests exist.

0.40, 95% CI = 0.34–0.48, P = 0.0143). In terms of individual complications, MIS appeared to have a positive effect in decreasing the complications of transfusion, wound infection, pelvic infection and abscess, lymphedema, intestinal obstruction, pulmonary embolism, deep vein thrombosis, and urinary tract infection. Furthermore, MIS had a negative effect in increasing the complications of cystotomy, bowel injury, subcutaneous emphysema, and fistula.

## Conclusions

Our meta-analysis demonstrates that MIS is superior to laparotomy, with fewer postoperative overall complications (wound infection, pelvic infection and abscess, lymphedema, intestinal obstruction, pulmonary embolism, and urinary tract infection). However, MIS is associated with a higher risk of intraoperative aggregate complications (cystotomy, bowel injury, and subcutaneous emphysema) and postoperative fistula complications.

## 1. Introduction

Being the fourth most common cancer among women, it has been estimated that there were approximately 528, 000 new cases of cervical cancer with 266, 000 deaths annually [1]. Until now, radical hysterectomy with an open abdominal approach was the predominant modality for the treatment of early cervical cancer [2]. After 1992, with the development of laparoscopic approach, minimally invasive surgery (MIS, i.e., laparoscopy or robotic surgery) for radical hysterectomy to treat cervical cancer has been accepted widely as a standard treatment for early-stage cervical cancer [3].

Surprisingly, the results of Laparoscopic Approach of the Cervix (LACC) clinical trial showed that minimally invasive radical hysterectomy was associated with lower rates of disease-free survival and overall survival compared with open surgery in 2018 [4]. After that, the open abdominal approach was defined as the "standard and recommended approach to radical hysterectomy" for cervical cancer by the National Comprehensive Cancer Network (NCCN) guidelines [5]. Therefore, discussing the surgical complications have to be done clarifying better the actual role of MIS and laparotomy in cervical cancer.

Till date, the advantages of MIS over laparotomy for management of gynecological diseases have been cited in the literature to included less blood loss, shorter hospital stay, and faster recovery [6–8]. Similarly, most previous studies on this subject also showed that robotic surgery has the advantages of providing a three-dimensional perspective and more accurate surgical positioning than laparotomy [9–11]. However, MIS was also associated with its complexity of operation, longer learning curve, and higher cost than laparotomy. Therefore, there is no good evidence of the overall advantage of one surgical modality over another, and it is uncertain whether MIS is justifiable in terms of safety and efficacy, due to the small sample sizes, the low-quality of previous studies, and the limited number of randomized controlled trials (RCTs).

As for complications, many previous studies showed that MIS and open radical hysterectomy (ORH) have no difference in terms of intraoperative and postoperative complications [12]. With further development of instruments and skills, several studies found that MIS was associated with lower rate of intraoperative and postoperative complications than laparotomy [13]. Unfortunately, till date, it is unclear whether the rates of individual complications in MIS

are also less than what are seen in laparotomy. Further emphasizing the severity of complications, which are a key factor in the evaluation of cervical cancer.

The aim of this meta-analysis was to compare the published rates of common intraoperative and postoperative complications between ORH and MIS in order to provide valid evidence for evaluating the advantages of different surgical procedures for managing cervical cancer.

## 2. Methods

### 2.1. Search strategy

A comprehensive, systemic search for articles was performed using the databases of PubMed, Embase, Cochrane library, and Web of science. We searched the articles in each database from the data of its inception until—February 2020. Search terms included a combination of synonyms and abbreviations relating to cervical cancer, laparoscopy, laparotomy, robotic surgery, and complication. All articles that met the eligibility criteria were assessed. The details of the search strategy are shown in S1 Table.

### 2.2. Selection criteria

Studies were included if they met the following criteria: (1) Patients were classified as stage IA-IIB (according to the 2018 International Federation of Gynecology and Obstetrics classification); (2) Subjects were females who underwent LRH, laparoscopic-assisted vaginal radical hysterectomy (LAVRH), RAH or ORH as primary treatment for cervical cancer; (3) The outcomes of complications in MIS and ORH were reported. Articles were excluded if they met the following criteria: (1) Patients received other treatments (radiation or concurrent chemoradiation therapy) before surgery; (2) The articles were case reports, reviews, meta-analysis, organizational guidelines, letters, expert opinions, or conference abstracts; (3) The studies had inadequate data for outcome assessment; (4) The articles had no outcomes of interest. (5) The published Articles were not in English.

### 2.3. Data extraction and quality assessment

Data were extracted into a standard form, and included information on the first author, publication year, country, participants' characteristics, study design, number of study participants, surgical approaches, and FIGO stage. Primary outcomes were intraoperative total complications and postoperative aggregate complication. Secondary outcomes were categorized into two groups (individual intraoperative and postoperative complications). Individual intraoperative complications included bladder damage, cystotomy, bowel injury, subcutaneous emphysema, nerve injury, ureteral injury, and vessel injury. Postoperative complications included wound infection, incisional hernia, pelvic infection and abscess, lymphedema, lymphocyst, intestinal obstruction, pulmonary embolism, deep vein thrombosis, and fistula. In this meta-analysis, we used the Newcastle-Ottawa scale to evaluate 39 studies and the Jadad scale to evaluate 1 study S2 and S3 Tables [14, 15]. Two reviewers independently evaluated and cross-checked the qualities of the included studies, as well as assessed the bias of the studies. Disagreements were discussed between two evaluators in order to reach a consensus and the third reviewer also provided the opinion.

### 2.4. Data synthesis and meta-analysis

This meta-analysis was conducted using Stata SE version 12.0 software (StataCorp, College Station, TX). We analyzed heterogeneity with the chi-square test, and P-value $< 0.10$ was used to establish statistical significance with $I^2$ test [16]. $I^2$ values $> 50\%$ were considered substantial

evidence of statistical heterogeneity. To estimate pooled odds ratio (OR) with 95% confidence interval (CI), a fixed-effects model was used in the absence of significant heterogeneity; the random-effects model was used in the presence of significant heterogeneity [17]. We evaluated the publication bias for each of the pooled study groups with a funnel plot. We carried out subgroup analysis based on the modalities of MIS (LAVRH, total laparoscopic radical hysterectomy (TLRH), and RRH) to assess the outcomes of different subgroups.

## 3. Results

A total of 40 studies were included in this analysis. The flowchart of the selection process is shown in Fig 1. The initial search retrieved 3,673 articles from the four databases. All articles were imported into Endnote for screening. After excluding duplicates, 1,887 articles were identified for the next step of screening. By reviewing titles and abstracts, 1,798 articles were removed for not meeting the selection criteria, and 89 articles were identified to be assessed for eligibility. Eventually, 40 studies were identified in the final analysis, and all of them were screened after reviewing the full text. We used the Newcastle-Ottawa scale to assess the quality of 39 studies and Jadad scale to assess 1 RCT, Table 1 shows the results of included studies.

The main characteristics of the 40 studies are shown in Table 1. The study designs were as follow: retrospective study (n = 31) [18–48], retrospective matched study (n = 6) [49–54], prospective cohort study (n = 2) [55, 56], and RCT (n = 1) [57]. Thirteen studies were conducted in Asia (China, Israel, Korea, Singapore, and Taiwan) [21, 35, 36, 38, 39, 41, 42, 45, 47, 52, 54–

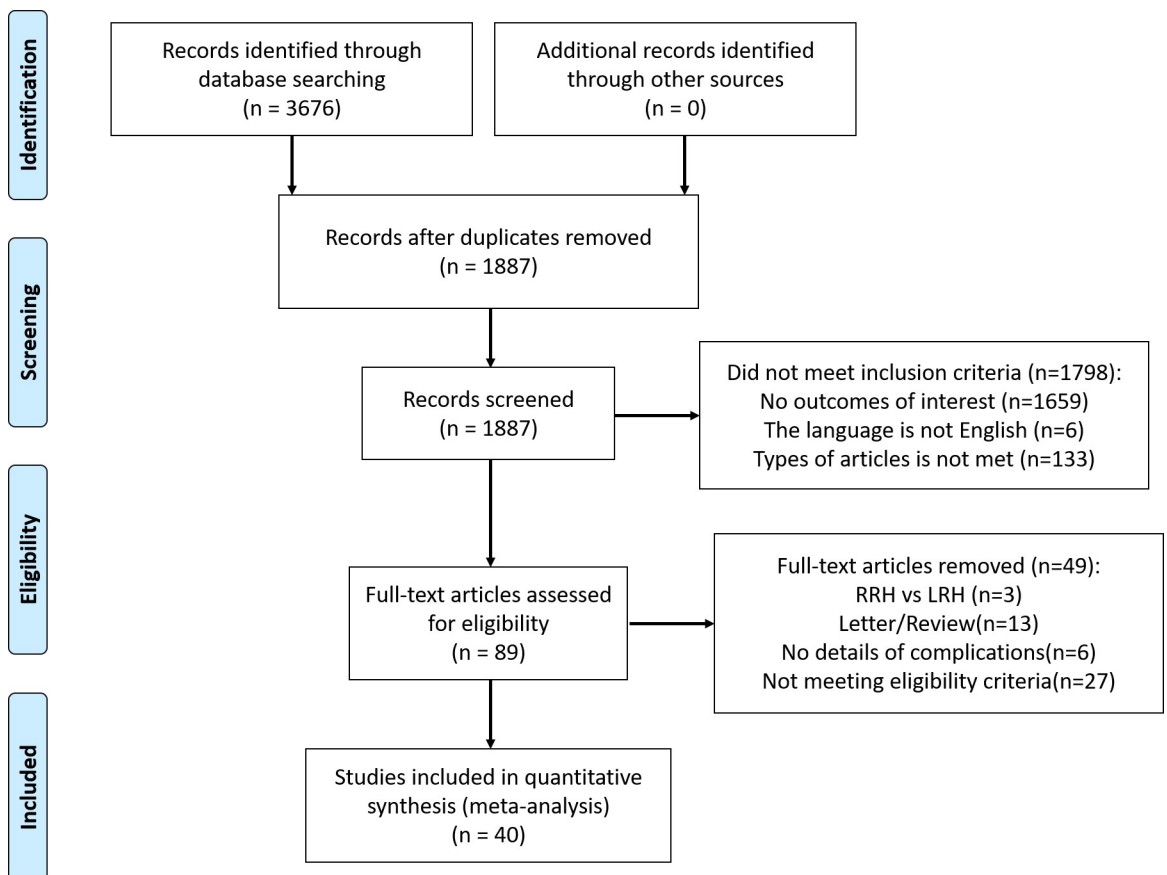

**Fig 1. Flow chart of study selection in this meta-analysis.**

**Table 1. Characteristics of the 40 studies included in the meta-analysis.**

| Study cohort | Year | Country | Study design | Approach | Number(N) | FIGO stage (N) | BMI[a] | Age[a](years) | Score[b] |
|---|---|---|---|---|---|---|---|---|---|
| Lee et al. | 2002 | China, Taiwan | prospective | LAVRH | 30 | IA-IB 30 | 54.4 ± 12.6 | 46.2(32–64) | 6 |
| | | | | ORH | 30 | IA-IB 30 | 56.3 ± 10.4 | 48.0(34–68) | |
| Steed et al. | 2004 | Canada | retrospective | LAVRH | 71 | IA-IB 71 | - | 43 (30–69) | 6 |
| | | | | ORH | 205 | IA-IB 205 | - | 44 (24–86) | |
| Sharma et al. | 2006 | England | retrospective | LAVRH | 35 | IA2–IIB 35 | - | 43.4(28–60) | 8 |
| | | | | ORH | 32 | IA2–IIB 32 | - | 42.8(28–66) | |
| Frumovitz et al. | 2007 | USA | retrospective | LRH | 35 | IA-IB 35 | 28.1(18.4–40.8) | 40.8(28.4–63.4) | 8 |
| | | | | ORH | 54 | IA-IB 54 | 28.2(17.4–46.4) | 42.5(27.3–68.3) | |
| Li et al. | 2007 | China | retrospective | LRH | 90 | IB-IIA 90 | - | 42 ± 9 | 6 |
| | | | | ORH | 35 | IB-IIA 35 | - | 44 ± 11 | |
| Morgan et al. | 2007 | Ireland | retrospective matched | LAVRH | 30 | IA–IB 30 | 25 (18.6–47) | 35 (25–54) | 6 |
| | | | | ORH | 30 | IA–IIB 30 | 24 (19.8–29.5) | 38 (20–63) | |
| Uccella et al. | 2007 | Italy | retrospective | LRH | 50 | IA2–IIA 50 | 23 (17.4–35) | 47 (24–78) | 7 |
| | | | | ORH | 48 | IA2–IIA 48 | 25 (19–43) | 53 (28–75) | |
| Zakashansky et al. | 2007 | USA | retrospective matched | LRH | 30 | IA1–IIA 30 | - | 48.3 ± 12.25 | 7 |
| | | | | ORH | 30 | IA1–IIA 30 | - | 46.6 ± 11.75 | |
| Boggess et al.[c] | 2008 | USA | retrospective | RRH | 51 | IA1–IIA 47 | 28.6 ± 7.2 | 47.4 ± 12.9 | 6 |
| | | | | ORH | 49 | IA2–IIA 49 | 26.1 ± 5.1 | 41.9 ± 11.2 | |
| Ko et al. | 2008 | USA | retrospective | RRH | 16 | IA1–IB1 16 | 27.6 ± 6.4 | 42.3 ± 7.9 | 5 |
| | | | | ORH | 32 | IA1–IIA 32 | 26.6 ± 5.9 | 41.7 ± 8.1 | |
| Estape et al. | 2009 | USA | retrospective | RRH | 32 | IB1-IB2 32 | 29.7 ± 3.2 | 55.0(33–83) | 7 |
| | | | | LRH | 17 | IA2-IB2 17 | 28.1 ± 4.8 | 52.8(37–83) | |
| | | | | ORH | 14 | IB1-IB2 14 | 29.5 ± 6.4 | 42.0(27–71) | |
| Maggioni et al. | 2009 | USA | retrospective | RRH | 40 | IA2–IIA 40 | 24.1 ± 5.5 | 44.1 ± 9.1 | 7 |
| | | | | ORH | 40 | IA2–IIA 40 | 23.6 ± 5.0 | 49.8 ± 14.1 | |
| Malzoni et al. | 2009 | Italy | retrospective | TLRH | 65 | IA1–IB1 65 | 26(19–35) | 40.5 ± 7.7 | 9 |
| | | | | ORH | 62 | IA1–IB1 62 | 29(19–35) | 42.7 ± 8.6 | |
| Papacharalabous et al. | 2009 | UK | retrospective | LAVRH | 14 | IA2–IB 14 | - | 38.6 ± 3.6 | 8 |
| | | | | ORH | 12 | IA2–IB 12 | - | 43.5 ± 12.9 | |
| Sobiczewski et al. | 2009 | Poland | retrospective | LRH | 22 | IA1–IB1 22 | - | 45.44 ± 9 | 8 |
| | | | | ORH | 58 | IA1–IIA 58 | - | 51.19 ± 12 | |
| Schreuder et al. [c] | 2010 | Netherlands | retrospective | RRH | 13 | IB1–IIB 13 | - | 43 (31–78) | 7 |
| | | | | ORH | 14 | IB1-IB2 14 | - | 46 (32–68) | |
| Lee et al. | 2011 | ROK | retrospective | LRH | 24 | IA2–IIa 24 | 23.4±3.55 | 48.4 ± 7.25 | 9 |
| | | | | ORH | 48 | IA2–IIa 48 | 23.9±4.7 | 50.2 ± 8.25 | |
| Sert et al. | 2011 | Norway | retrospective | RRH | 35 | IA1–IB1 35 | 25.4±4.36 | 44.1 ± 10.5 | 9 |
| | | | | LRH | 7 | IA1–IB1 7 | 22.5±1.84 | 45.0 ± 12.9 | |
| | | | | ORH | 26 | IA1–IB1 26 | 25±3.0 | 44.8 ± 11.8 | |
| Taylor et al. | 2011 | USA | retrospective | LAVRH | 9 | IA2–IB1 9 | 26.3 (20.6–36.1) | 41.4 (31–60) | 7 |
| | | | | ORH | 18 | IA2–IB1 18 | 26.9 (17–38.3) | 41.1 (25–61) | |
| Gortchev et al. | 2012 | Bulgaria | retrospective | RRH | 73 | - | - | 46.0 ± 11.2 | 8 |
| | | | | LAVRH | 46 | - | - | 42.5 ± 9.9 | |
| | | | | ORH | 175 | - | - | 49.0 ± 11.0 | |
| Nam et al. | 2012 | Korea | retrospective matched | LRH | 263 | IA2–IIA 263 | - | - | 8 |
| | | | | ORH | 263 | IA2–IIA 263 | - | - | |
| Park et al. | 2012 | Korea | retrospective | LRH | 54 | IA2–IIA2 54 | 31.8 ± 1.39 | 49.4 ± 11.5 | 7 |
| | | | | ORH | 112 | IA2–IIA2 112 | 31.7 ± 1.5 | 52.1 ± 11.8 | |

(*Continued*)

**Table 1.** (Continued)

| Study cohort | Year | Country | Study design | Approach | Number(N) | FIGO stage (N) | BMI[a] | Age[a](years) | Score[b] |
|---|---|---|---|---|---|---|---|---|---|
| Lim et al. | 2013 | Singapore | prospective | LRH | 18 | IA1-IIA 18 | 22.9 (16.0–33.7) | 48 (30–65) | 9 |
| | | | | ORH | 30 | IA1-IIA 30 | 22.4 (17.9–33.9) | 47 (33–67) | |
| Park et al. | 2013 | Korea | retrospective | LRH | 115 | IB2-IIA2 115 | 23.1 (15.6–34.8) | 48.5 (25–77) | 8 |
| | | | | ORH | 188 | IB2-IIA2 188 | 23.7 (17.6–34.7) | 48.1 (25–84) | |
| BoganI et al. | 2014 | Italy | retrospective | LRH | 65 | IA2-IIB 65 | 25.1 ± 5.2 | 48.9 ±13.5 | 9 |
| | | | | ORH | 65 | IA2-IIB 65 | 25.9 ± 6.1 | 50.9 ± 14 | |
| Chen et al. | 2014 | Taiwan | retrospective | RRH | 24 | IA-IIB 24 | 24.4 ± 4.9 | 53.7 ± 15.3 | 8 |
| | | | | LRH | 32 | IA-IIB 32 | 23.2 ± 3.4 | 51.2 ± 11.9 | |
| | | | | ORH | 44 | IA-IIB 44 | 24.9 ± 4.6 | 51.9 ± 11.3 | |
| Yin et al. | 2014 | China | retrospective | LRH | 22 | IA2–IIA 22 | - | 44 ± 1.5 | 6 |
| | | | | ORH | 23 | IA2–IIA 23 | - | 46 ± 2.3 | |
| Asciutto et al. | 2015 | Sweden | retrospective | RRH | 64 | IA2–IIA 64 | 27.0 ± 6.1 | 45.4 ± 13.6 | 6 |
| | | | | ORH | 185 | IA2–IIA 178 | 25.7 ± 4.7 | 45.7 ± 13.0 | |
| Ditto et al. | 2015 | Italy | retrospective matched | LRH | 60 | IA2–IB1 60 | 24.3 ± 2.9 | 46 (29–79) | 9 |
| | | | | ORH | 60 | IA2–IB1 60 | 24.0 ± 4.3 | 45.5 (15–78) | |
| Xiao et al. | 2015 | China | retrospective | LRH | 106 | IA-IIB 106 | 23.8 ± 3.9 | 43.7 ± 9.3 | 8 |
| | | | | ORH | 48 | IA-IIB 48 | 24.7 ± 3.8 | 45.7 ± 11.3 | |
| Park et al. | 2016 | Korea | retrospective | LRH | 186 | IA2–IIA1 186 | 23.69 (17.1–34.9) | 45.3 (27–71) | 7 |
| | | | | ORH | 107 | IA2–IIA1 107 | 23.58 (17.1–35.9) | 47.3 (28–73) | |
| Shah et al. | 2017 | USA | retrospective | RRH | 109 | IA1-IB2 109 | 27.9 (17.6–51.6) | 45.2 (25–84) | 7 |
| | | | | ORH | 202 | IA1-IB2 202 | 29.1 (18.3–55.7) | 45.4 (19–88) | |
| Corrado et al. | 2018 | Italy | retrospective | RRH | 88 | IB1 88 | 23.3 (18–47.6) | 46 (27–77) | 8 |
| | | | | LRH | 152 | IB1 152 | 23.5 (17–35) | 45 (23–78) | |
| | | | | ORH | 101 | IB1 101 | 24.8 (18–51) | 50 (28–76) | |
| Guo et al. | 2018 | China | retrospective | LRH | 412 | IA-IIA 412 | 22.81 (14.3–35.6) | 44.19 (25–76) | 7 |
| | | | | ORH | 139 | IA-IIA 139 | 23.19 (13.8–36.6) | 40.52 (23–62) | |
| Bogani et al. [c] | 2019 | Italy | Retrospective matched | LRH | 35 | IB1-IIA 23 | 22.9 ± 4.0 | 41.1 ± 6.9 | 7 |
| | | | | ORH | 35 | IB1-IIA 24 | 20.1 ± 9.3 | 44.1 ± 12.7 | |
| Matanes et al. | 2019 | Israel | retrospective | RRH | 74 | IA1-IIA 74 | 26.4(18.2–42.1) | 48(29–77) | 8 |
| | | | | ORH | 24 | IA1-IIA 24 | 26.2(20.6–38.5) | 47(24–69) | |
| Piedimonte et al. | 2019 | Canada | Retrospective | RRH | 749 | - | - | - | 6 |
| | | | | ORH | 2584 | - | - | - | |
| Yuan et al. | 2019 | China | Retrospective matched | LRH | 99 | IIA2-IIA2 99 | 44.56 ± 7.60 | 43.58 ± 8.86 | 9 |
| | | | | ORH | 99 | IIA2-IIA2 99 | 24.56 ± 1.50 | 44.56 ± 7.60 | |
| Pahisa et al. | 2010 | Spain | Retrospective | LAVRH | 67 | IA2-IIA 67 | 25.4 ± 1.1 | 51 (29–75) | 7 |
| | | | | ORH | 23 | IA2-IIA 23 | 27.2 ± 2.5 | 48 (31–67) | |
| Campos et al. | 2013 | Brazil | RCT | LRH | 16 | IA2–IB 16 | - | 36.19 ± 9.78 | 5 |
| | | | | ORH | 14 | IA2–IB 14 | - | 39.64 ± 6.23 | |

ORH: Open radical hysterectomy, LRH: Laparoscopic radical hysterectomy, RRH: Robotic radical hysterectomy, LAVRH: Laparoscopic-assisted vaginal radical hysterectomy, RCT: Randomized controlled trial

a: Mean, median or unknow.

b: Jadad scale: score: 1~3, indicating low quality study; score: 4~7, indicating high quality study. Newcastle-Ottawa scale: score≤5, indicating high risk of bias; score>5, indicating low risk of bias.

c: These studies including other FIGO stages of cervical cancer.

56], ten in North America (Canada, and USA) [18, 20, 23–26, 33, 43, 48, 51], sixteen in Europe (UK, Ireland, Poland, Netherlands, Federal Republic of Germany, Norway, Bulgaria, Italy, Sweden, and Spain) [19, 22, 27–32, 34, 37, 40, 44, 46, 49, 50, 53], and one study in South America (Brazil) [57]. In all, we identified 9003 patients in the pooled analysis: 2277 patients had LRH, 1,368 patients had RRH and 5358 patients had ORH (we compared 1,368 patients who underwent RRH vs 3,490 patients who underwent ORH, and 2277 patients who underwent LRH vs 2,228 patients who underwent ORH). As shown in Table 1, 8 studies compared RRH with ORH [23, 24, 26, 29, 39, 42, 46, 47], 25 studies compared LRH with ORH [18–22, 27, 28, 30, 32, 34–36, 38, 40, 41, 44, 45, 48–57], and 5 studies compared all 3 surgical approaches [25, 31, 33, 37, 43].

## 3.1 MIS vs ORH

**3.1.1 Primary outcomes.**  We show the results of intraoperative aggregate complications and postoperative overall complications between MIS and ORH in Fig 2, respectively. For intraoperative complications, the incidence of intraoperative complications in MIS (121/3459) were statistically higher than ORH (102/5174), and the risk of intraoperative complications (OR = 1.41, 95% CI = 1.07–1.86, P<0.05) in MIS was higher compared with ORH. In terms of postoperative complications, MIS was associated with significantly lower risk of postoperative complications (OR = 0.40, 95% CI = 0.34–0.48, P = 0.0143) compared with ORH. There was no heterogeneity in studies of intraoperative aggregate complications ($I^2$ = 0%, P = 0.748). However, we found that the studies of postoperative overall complications were associated with high heterogeneity ($I^2$ = 51%, P<0.01). The result of publication bias was shown in Fig 3, the funnel plot was nearly symmetric on both sides, so there was no publication bias in the results of intraoperative aggregate complications and postoperative overall complications.

**3.1.2 Secondary outcomes.**  In order to determine the source of difference, we analyzed the results of individual intraoperative and postoperative complications in Table 2, respectively. For intraoperative complications, there were no significant differences between MIS and ORH in the bladder damage, nerve injury, ureteral injury, or vessel injury, with ORs of 1.28 (95% CI = 0.75–2.19, P = 0.3), 0.51 (95% CI = 0.14–1.93, P = 0.343), 1.05 (95% CI = 0.61–1.76, P = 0.959), 1.01 (95% CI = 0.59–1.73, P = 0.753), respectively. However, MIS was associated with increased risk of cystotomy (OR = 2.27, 95% CI = 1.23–4.20), bowel injury (OR = 2.15, 95% CI = 0.95–4.89), subcutaneous emphysema (OR = 4.36, 95% CI = 0.94–20.29) in comparison with ORH. In terms of postoperative complications, there were comparable in the risk of incisional hernia (OR = 0.93, 95% CI = 0.34–2.51, P = 0.803) and lymphocyst (OR = 0.73, 95% CI = 0.46–1.15, P = 0.123) between MIS and ORH. Comparing to ORH, MIS was associated with significantly lower risks of wound-infection (OR = 0.15, 95% CI = 0.08–0.28, P<0.01), pelvic infection and abscess (OR = 0.40, 95% CI = 0.26–0.63, P<0.01), lymphedema (OR = 0.48, 95% CI = 0.24–0.98, P = 0.03), intestinal obstruction (OR = 0.30, 95% CI = 0.21–0.43, P<0.01), pulmonary embolism (OR = 0.36, 95% CI = 0.09–1.48, P = 0.025), deep vein thrombosis (OR = 0.56, 95% CI = 0.35–0.88, P = 0.01), and urinary tract infection (OR = 0.56, 95% CI = 0.34–0.91, P = 0.013). However, the risk of fistula (OR = 1.69, 95% CI = 0.02–2.79, P = 0.011) was significant increased in the MIS group than in ORH.

## 3.2 Subgroup analysis

The subgroup analysis compared intraoperative complications and postoperative complications among the three types of MIS, as shown in Table 3. For intraoperative aggregate complications, compared to ORH, the risks of complications were not increased in RRH (OR = 1.11, 95% CI = 0.62–2.01, P = 0.11) and TLRH (OR = 1.34, 95%CI = 0.94–1.93, P = 0.722), whereas

# Intraoperative complications

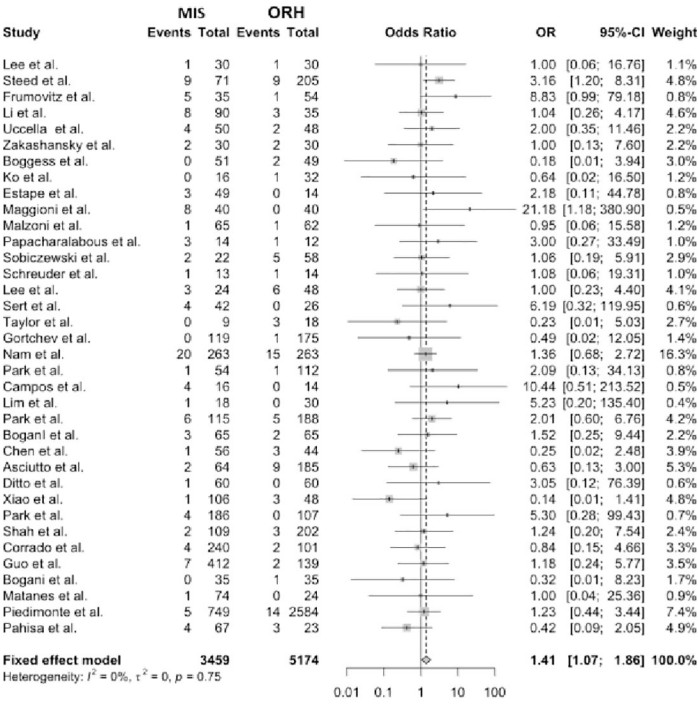

# Postoperative complications

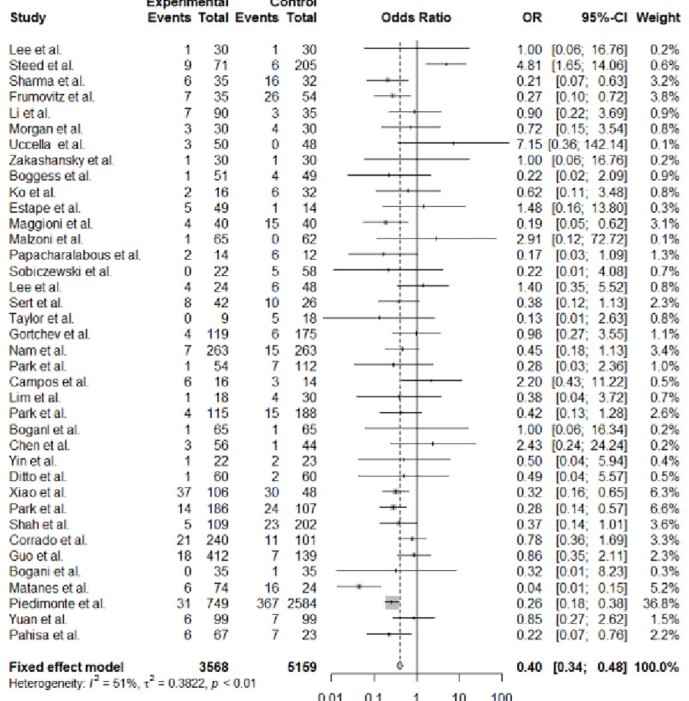

**Fig 2. Forest plots of intraoperative and postoperative complications between Minimally Invasive Surgical (MIS) and Open Radical Hysterectomy (ORH).** OR, odds ratio.

it was higher in LAVRH (OR = 2.27, 95%CI = 1.02–5.04, P = 0.044). For postoperative overall complication, the risk in LAVRH (OR = 0.71, 95%CI = 0.26–1.93, P = 0.506) was not statistically different from that of ORH. However, RRH (OR = 0.42, 95%CI = 0.26–0.68, P<0.01) and TLRH (OR = 0.58, 95%CI = 0.45–0.74, P<0.01) was associated with a reduced risk of postoperative complication when compared with ORH. In a stratified analysis (S4 Table), in an attempt to further determine the difference in fistula complications, we also analyzed complications with different types of fistula, including vesicovaginal, rectovaginal, ureterovaginal and urinary fistula, with ORs of 1.55 (95%CI = 0.59–4.06, P = 0.376), 2.88 (95%CI = 0.44–18.70, P = 0.269), 1.60 (95%CI = 0.59–4.34, P = 0.353), and 1.25 (95%CI = 0.53–2.97, P = 0.612) respectively. Interestingly, there was no significant difference in risk of the individual fistula types between MIS and ORH.

## 4. Discussion

This study assessed most comprehensive results of complications of cervical cancer surgeries and evaluated the safety of different surgical strategies. The rates of perioperative complications will become a key factor of importance in comparing surgical modalities for managing cervical cancer. We aimed to provide a basis for the selection of optimal surgical methods, as well as offer new opinions for actual role of MIS in cervical cancer.

Our meta-analysis indicated that the overall risk of intraoperative complications was increased with MIS than with ORH. Patients accepted to MIS experienced almost 2 times the risk of intraoperative complications compared with patients accepted to ORH. There were no significant differences in risk for intraoperative complications including bladder damage, nerve injury, ureteral injury, and vessel injury among individual intraoperative complications. However, MIS group was associated with higher risk in complications of cystotomy, bowel injury, and subcutaneous emphysema in comparison to ORH. This finding was consistent with previous studies. The differences in bowel injury between MIS and ORH can be explained by the use of surgical instruments such as a trocar and Veress needle during radical hysterectomy. Previous studies have shown that the majority of bowel injuries occurred during laparoscopy using a Veress needle or trocar placement [58, 59]. The subcutaneous emphysema was the unique complications in MIS, many risk factors will lead to it during MIS including increased intra-abdominal pressure, total gas volume, and gas flow rate [60].

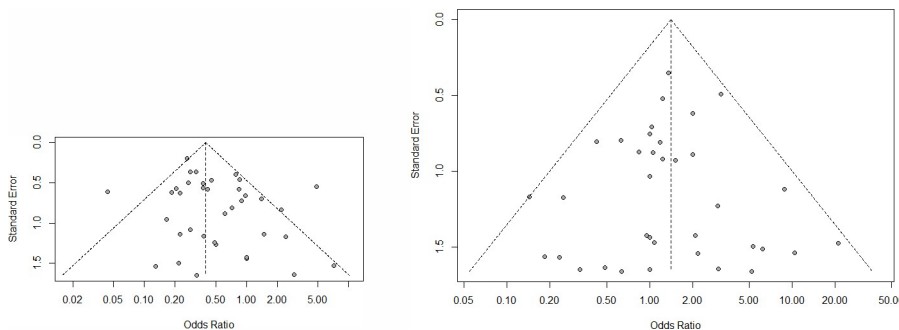

**Fig 3. Funnel plot of studies evaluating the publication bias of intraoperative and intraoperative complications between MIS and ORH.** (A). intraoperative complications. (B). postoperative complications.

**Table 2. Meta-analysis estimates of individual complications between MIS and ORH.**

| Category | MIS | ORH | OR (95% CI) | P value | $I^2$(%) |
|---|---|---|---|---|---|
| Transfusion | 301/2490 | 494/4408 | 0.34[0.22,0.53] | <**0.001** | 72.3 |
| **Intraoperative complications** | | | | | |
| Bladder damage | 25/2279 | 24/4009 | 1.28[0.75,2.19] | 0.3 | 0 |
| Cystotomy | 32/586 | 14/677 | 2.27[1.23,4.20] | **0.002** | 0 |
| Bowel injury | 12/1479 | 8/3449 | 2.15[0.95,4.89] | **0.041** | 0 |
| Subcutaneous emphysema | 7/246 | 0/207 | 4.36[0.94,20.29] | **0.008** | 0 |
| Nerve injury | 2/1181 | 5/802 | 0.51[0.14,1.93] | 0.343 | 0 |
| Ureteral injury | 22/2519 | 24/4520 | 1.05[0.61,1.76] | 0.959 | 0 |
| Vessel injury | 21/2328 | 27/4112 | 1.01[0.59,1.73] | 0.753 | 0 |
| **Postoperative complications** | | | | | |
| Wound infection | 5/1380 | 104/3277 | 0.15[0.08,0.28] | <**0.001** | 0 |
| Incisional hernia | 7/898 | 7/811 | 0.93[0.34,2.51] | 0.803 | 0 |
| Pelvic infection and abscess | 30/1713 | 78/3396 | 0.40[0.26,0.63] | <**0.001** | 39.9 |
| Lymphedema | 13/791 | 19/619 | 0.48[0.24,0.98] | **0.03** | 0 |
| Lymphocyst | 40/1614 | 35/1194 | 0.73[0.46,1.15] | 0.123 | 8.4 |
| Intestinal obstruction | 37/2490 | 281/4070 | 0.30[0.21,0.43] | <**0.001** | 0 |
| Pulmonary embolism | 0/508 | 7/558 | 0.36[0.09,1.48] | **0.025** | 0 |
| Deep vein thrombosis | 31/2289 | 78/3886 | 0.56[0.35,0.88] | **0.01** | 0 |
| Fistula | 38/2203 | 17/1904 | 1.69[0.02,2.79] | **0.011** | 0 |
| Urinary tract infection | 33/764 | 44/799 | 0.56[0.34,0.91] | **0.013** | 3 |

OR: Odds ratio; CI: Confidence interval; MIS: Minimally invasive surgery; ORH: Open radical hysterectomy

Regarding postoperative complications, our meta-analysis found that MIS was associated with significantly lower risk of postoperative overall complications compared with ORH. In individual postoperative complications, incisional hernia and lymphocyst had no differences between MIS and ORH. MIS was superior to ORH in terms of wound infection, pelvic infection and abscess, lymphedema, intestinal obstruction, pulmonary embolism, and urinary tract infection, whereas the risk of fistula complications was significantly increased, with MIS compared to ORH. Interestingly, in a stratified analysis of fistula complications, we found that there were no significant differences in risk for four types of fistula complications. Possible reasons for this result including individual fistula complication had small sample size and

**Table 3. The subgroup analysis of laparoscopic types between MIS and ORH in intraoperative and postoperative overall complications.**

| Category | Laparoscopic type | Study | OR (95% CI) | P value | $I^2$(%) |
|---|---|---|---|---|---|
| **Intraoperative complications** | | | | | |
| | TLRH | 23 | 1.34[0.94,1.93] | 0.11 | 0 |
| | LAVRH | 5 | 2.27[1.02,5.04] | **0.044** | 0 |
| | RRH | 13 | 1.11[0.62,2.01] | 0.722 | 0 |
| **Postoperative complications** | | | | | |
| | TLRH | 25 | 0.58[0.45,0.74] | <**0.01** | 0 |
| | LAVRH | 7 | 0.71[0.26,1.93] | 0.506 | 58.5 |
| | RRH | 11 | 0.42[0.26,0.68] | <**0.01** | 45.3 |

OR: Odds ratio; CI: Confidence interval; TLRH: Total laparoscopic radical hysterectomy; RRH: Robotic radical hysterectomy; LARVH: Laparoscopic assisted radical vaginal hysterectomy.

excessive weight of included studies biased the results [61]. Although there were no significant differences in the risks of vesicovaginal, rectovaginal, ureterovaginal, and urinary fistula between MIS and ORH, the incidence rates of these four types of fistula complications in MIS were higher than that of ORH. This finding was worthy of our attention.

Taken together, the surgeon proficiency may be a factor in determining the rates of complications. Regrettably, this meta-analysis was not able to provide a comparison between surgeons. Furthermore, the learning curve could play an important role in complications between different surgical modalities, and MIS was associated with a longer learning curve than ORH because of the complexity of surgical procedure, and also might have influenced complication rates [62, 63]. The use of surgical instruments was related to viscus injuries, which may be caused by thermal injury, due to the high temperature of the surgical instruments resulting in the damage of submucosal or deeper tissues of the bladder, intestines, and bowel. Previous studies have evaluated the thermal injury of bowel in laparoscopic approach [62]. It must be taken that thermal injury was an inherent risk of the technique during radical hysterectomy, and therefore surgeon should pay attention to this issue. Overall, these factors were associated with the incidence of intraoperative and postoperative complications.

Concerning the subgroup meta-analysis of surgical modalities, intraoperative complication rate increased in the course of LAVRH, as well as there were no differences in TLRH and RRH. This finding is consistent with that of previous meta-analyses. The requirement for refinement of LAVRH is very high due to the complex pelvic floor anatomy in females. In the vaginal approach, the ureters and bladder are identified by traction on the uterus after the ligament around the uterus is isolated and cut [64], and urinary tract trauma is a clear risk during LAVRH. With time, laparoscopy is continually evolving with the improvement in surgical skills, instruments, and learning curve, and these improvements may be partly responsible for reduction in intraoperative complication over time [63]. For postoperative aggregate complications, both RRH and TLRH were associated with lower risk compared to ORH. These results were validated in previous studies, Park et al. compared the complications of three approaches, RRH had a positive effect in reducing overall complications than ORH for cervical cancer patients [65]. For LAVRH group, the high heterogeneity and the small sample size could bias the results of postoperative complications. In the future, we need more high-quality cohort studies to evaluate and compare the risk of postoperative complications between MIS and ORH.

There are limitations to this meta-analysis. First, included studies were primarily non-randomized studies, which could not provide high-quality evidence. Furthermore, our study did not include single-arm studies, which can lead to the bias of the result. Additionally, differences in patients' characteristics between different surgical cohorts may lead to highly heterogeneous outcomes in studies and affect the results of the pooled analysis. The statistical methods could not fully diminish these differences. Second, the difference of surgeons in these articles were not reported including the level of experience in surgeons and types of surgeons, these factors could affect the surgical outcomes as time went by. The additional morbidities of patients in these studies were not involved, these factors could contribute to the bias of results. Third, most studies included in this meta-analysis did not use standardized methods of classifying complications, such as the Clavien-Dindo classification system, and the final results may be affected by these differences in the reporting of complications. Among all included studies, only one adopted the Clavien-Dindo classification system of complications [39]. Forth, during the extraction of complication data, many studies revealed that patients had undergone cesarean section or previous abdominal surgery and had severe adhesions in the past, alluding to the fact that the success of laparoscopy will be affected by adhesions. Therefore, the incidence of complications ultimately may interfere with the results and may be a cause of bias.

## 5. Conclusion

Our meta-analysis demonstrates that MIS is superior to laparotomy, with fewer postoperative overall complications (wound infection, pelvic infection and abscess, lymphedema, intestinal obstruction, pulmonary embolism, and urinary tract infection). However, MIS is associated with a higher risk of intraoperative aggregate complications (cystotomy, bowel injury, and subcutaneous emphysema) and postoperative fistula complications. In the future, high-quality prospective studies and RCTs are needed to provide sufficient evidence for evaluating the pros and cons of using MIS to treat cervical cancer.

## Supporting information

**S1 Checklist. PRISMA 2009 checklist.**
(DOC)

**S1 Fig. PRISMA 2009 flow diagram.**
(DOC)

**S1 Table. Detailed search strategy.**
(DOC)

**S2 Table. Quality assessment of the included studies according to modified NOS score.**
(DOC)

**S3 Table. Quality assessment of the included studies according to modified Jadad score.**
(DOC)

**S4 Table. The subgroup analysis of fistula types between MIS and ORH.**
(DOC)

## Author Contributions

**Conceptualization:** Yongjun Wang.

**Data curation:** Yilin Li.

**Funding acquisition:** Yongjun Wang.

**Investigation:** Qingduo Kong, Hongyi Wei.

**Methodology:** Yilin Li, Yongjun Wang.

**Supervision:** Hongyi Wei.

**Visualization:** Qingduo Kong.

**Writing – original draft:** Yilin Li.

**Writing – review & editing:** Yilin Li, Yongjun Wang.

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
