## [Decision Letter · Decision Letter 0]

26 Feb 2021

PONE-D-20-33497

Complications of surgical treatments for early-stage cervical cancer: a systematic review and meta-analysis

PLOS ONE

Dear Dr. Wang,

Thank you for submitting your manuscript to PLOS ONE. After careful consideration, we feel that it has merit but does not fully meet PLOS ONE’s publication criteria as it currently stands. Therefore, we invite you to submit a revised version of the manuscript that addresses the points raised during the review process.

We look forward to receiving your revised manuscript.

Kind regards,

Antonio Simone Laganà, M.D., Ph.D.

Academic Editor

PLOS ONE

Additional Editor Comments:

The topic of the manuscript is interesting. Nevertheless, the reviewers raised several concerns: considering this point, I invite authors to perform the required major revisions.

Journal Requirements:

"The study was funded by the Capital’s Funds for Health Improvement and Research in China. (2020-2-8022)"

"YES - Specify the role(s) played"

Reviewers' comments:

Reviewer's Responses to Questions

**Comments to the Author**

1. Is the manuscript technically sound, and do the data support the conclusions?

Reviewer #1: Partly

Reviewer #2: Yes

2. Has the statistical analysis been performed appropriately and rigorously? 

Reviewer #1: No

Reviewer #2: Yes

3. Have the authors made all data underlying the findings in their manuscript fully available?

Reviewer #1: Yes

Reviewer #2: Yes

4. Is the manuscript presented in an intelligible fashion and written in standard English?

Reviewer #1: No

Reviewer #2: Yes

5. Review Comments to the Author

Reviewer #1: I was pleased to revise the manuscript entitled “Complications of surgical treatments for early-stage cervical cancer: a systematic review and meta-analysis” (Manuscript Number: PONE-D-20-33497).

In my honest opinion, the topic is interesting enough to attract the readers’ attention. Nevertheless, authors should clarify different points.

In general, the Manuscript may benefit from some major revisions, as suggested below:

- All the text needs a minor language revision in order to improve readability, some typos and grammatical errors.

- Title. I would suggest revising the title to provide more details regarding the review aim. If you are focusing on a comparison between MIS and open surgery, it have to be clear in the title.

- Abstract. I would suggest improving the method section of the abstract providing the inclusion criteria following the PICO system. The present form is too generic.

- The introduction regarding the surgical approach for cervical cancer is not update. After the LACC trial, it was clearly demonstrated that MIS for cervical cancer is associated with worse overall survival. Therefore, discussing the surgical morbidity have to be done clarifying better the actual role of MIS in cervical cancer.

- Methods. Search strategy and study selection should be improved. Line 99 is conflicting with lines 111-112. I would suggest reporting the study selection following the PICO system.

- Methods. Was a third investigator involved in the case of disagreement.

- Methods. How were the type of complications defined?

- Results, I would suggest providing some data regarding the absolute risk difference and not only the OR. If it is not possible in results. I would suggest discussing this point in the discussion. A change in risk from 10 to 20% and from 0.01 to 0.02 have the same OR but completely different clinical meaning.

- Data regarding the pooled cumulative incidence of complications should be provided.

- Jadad scale in Table 1 is not reported in the study’s methods.

- Lines 314-317. I would suggest provided references supporting the statements.

- Regarding study limitations, how complications were classified and standardized for this review should be clearly reported to allow reproducibility.

- I would suggest discussing better the concept regarding the pooling of studies with completely different study designs.

- The main concern regarding the current review is the background, which is completely out of the more recent literature regarding surgical approach for cervical cancer. This required extensive improvement. Refer to: PMID: 30380365; PMID: 32320800

Reviewer #2: The title of this manuscript does not well represent the topic—this is specifically a comparison of complications between minimally invasive and open surgical treatments for cervical cancer.

Why did the authors choose to focus only on studies that compare the two (or three) surgical methods and exclude case series? Given that only one study was randomized, it seems that they would have had a much larger number of articles without losing substantial rigor.

It would be helpful to ground more general readers into the stage at which women would be candidates for surgical treatment, and how that may differ in LMICs compared to HICs where there are more resources available for radiation.

Line 87: The aim of the meta-analysis was to compare the published rates of common intra-operative and post-operative complications.

Was there a reason that the authors did not abstract the disease stage? Was this not presented in the published studies?

Line 169: What does BMI mean in this context? Do you mean that in some studies, median BMI was reported vs mean vs no BMI description?

Lines 213-217: This needs to be worded more clearly, as the authors are stating that there is an increased odds of aggregate fistula formation, but when disaggregated, the odds of formation of individual fistula types were not increased, likely due to the small numbers in each category. The word “overall” in line 216 is confusing.

Discussion:

The second sentence in the discussion states that the authors aimed to offer new opinions for prognostic indicators in MIS—this was not described in the methods and none of the results would suggest prognostic indicators were evaluated.

Line 281: would change that sentence to read that intestinal obstruction is often related to abdominal adhesions, which are more likely to form after abdominal surgery. I would avoid the phrase “indirectly proved.”

Lines 297-302: There was no difference in fistula rates between ORH and MIS. Given the low numbers of fistulas seen overall, I would avoid making the over-conclusions that the authors may be doing in lines 298-302.

I would suggest that the entire paragraph 284-302 can be combined with the preceding two paragraphs, preferably the urinary complication paragraph. The second sentence, describing the incidence of complications as depending on the surgeon’s skills, is general to all of the complications, not just fistula formation.

Additional limitations that were not described include the lack of data on the surgeon level or years of experience, and additional patient morbidities (in addition to prior surgeries).

Table 2 needs a more descriptive title.

Figure 1: Include the reasons that articles were excluded and the #s per reason at the abstract and full-text stage in the flowchart.

6. PLOS authors have the option to publish the peer review history of their article (what does this mean?). If published, this will include your full peer review and any attached files.

Reviewer #1: No

Reviewer #2: No

---

## [Author Response · Author response to Decision Letter 0]

23 Apr 2021

We have solved these questions in the article and provided the latter of "Response to Reviewers"

---

## [Decision Letter · Decision Letter 1]

31 May 2021

Comparison of the complications between minimally invasive surgery and open surgical treatments for early-stage cervical cancer: a systematic review and meta-analysis

PONE-D-20-33497R1

Dear Dr. Wang,

We’re pleased to inform you that your manuscript has been judged scientifically suitable for publication and will be formally accepted for publication once it meets all outstanding technical requirements.

Kind regards,

Antonio Simone Laganà, M.D., Ph.D.

Academic Editor

PLOS ONE

Additional Editor Comments (optional):

Authors performed the required corrections, which were positively evaluated by the reviewers. I am pleased to accept this paper for publication.

Reviewers' comments:

Reviewer's Responses to Questions

**Comments to the Author**

1. If the authors have adequately addressed your comments raised in a previous round of review and you feel that this manuscript is now acceptable for publication, you may indicate that here to bypass the “Comments to the Author” section, enter your conflict of interest statement in the “Confidential to Editor” section, and submit your "Accept" recommendation.

Reviewer #1: All comments have been addressed

2. Is the manuscript technically sound, and do the data support the conclusions?

Reviewer #1: Yes

3. Has the statistical analysis been performed appropriately and rigorously? 

Reviewer #1: Yes

4. Have the authors made all data underlying the findings in their manuscript fully available?

Reviewer #1: Yes

5. Is the manuscript presented in an intelligible fashion and written in standard English?

Reviewer #1: Yes

6. Review Comments to the Author

Reviewer #1: In the new version of the manuscript, authors addressed all recommended revisions, and I appreciated the manuscript improvement.

7. PLOS authors have the option to publish the peer review history of their article (what does this mean?). If published, this will include your full peer review and any attached files.

Reviewer #1: No

---

## [Editor Report · Acceptance letter]

21 Jun 2021

PONE-D-20-33497R1 

Comparison of the complications between minimally invasive surgery and open surgical treatments for early-stage cervical cancer: a systematic review and meta-analysis 

Dear Dr. Wang:

I'm pleased to inform you that your manuscript has been deemed suitable for publication in PLOS ONE. Congratulations! Your manuscript is now with our production department. 

Kind regards, 

on behalf of

Dr. Antonio Simone Laganà 

Academic Editor

PLOS ONE